# WORSE TOGETHER: UNDERSTANDING THE BRITTLENESS OF MULTIMODAL MODELS ON RARE CONCEPT PAIRS

## ABSTRACT

Multimodal models are being deployed in real-world settings where rare or unseen combinations of objects during pretraining are bound to appear at test time. Understanding how these models generalize to rare combinations of concepts is thus an important robustness problem. In this paper, we investigate how the pairwise co-occurrence of concepts in the pretraining dataset impacts CLIP and large multimodal model (LMM) performance on uncommon concept pairs. We measure concept co-occurrence with pointwise mutual information (PMI), which corrects for the correlation between single and paired concept frequencies. We show a strong correlation between PMI in the CLIP pretraining data and zero-shot accuracy in CLIP models trained on LAION-400M ($r = 0.97$ and 14% accuracy gap between images in the top and bottom 5% of PMI values), and demonstrate that a simple PMI-based image edit can induce an accuracy drop of up to 10% on real images edited to contain low PMI pairs. We additionally find that this behavior in CLIP transfers to LMMs built on top of CLIP ($r = 0.70$ for TextVQA, $r = 0.62$ for VQAv2). Finally, we demonstrate that fine-tuning CLIP with augmented data covering a broad range of PMI values is a promising strategy to improve robustness on rare concept pairs.

## 1 INTRODUCTION

Contrastive image-text encoders such as CLIP (Radford et al., 2021; Cherti et al., 2023) are a crucial component of large multimodal models (LMMs) (Achiam et al., 2023; Liu et al., 2023a; Deitke et al., 2024; Awadalla et al., 2023), which have seen widespread adoption on a diverse array of vision-language tasks. A hallmark of CLIP is its strong zero-shot accuracy on challenging datasets, such as ImageNet-R and ObjectNet (Taori et al., 2020; Hendrycks et al., 2020; Barbu et al., 2019), leading to a perception of broad robustness (Fang et al., 2022; Li et al., 2023b; Xue et al., 2023; Mayilvahanan et al., 2024).

Recent work shows that CLIP exhibits higher zero-shot accuracy on examples involving common visual concepts in pretraining (Udandarao et al., 2024; Parashar et al., 2024). However, real-world images contain combinations of concepts that are inevitably rare or unseen in the pretraining dataset. The role of concept combinations in CLIP pretraining remains largely unclear – for instance, how does accuracy vary when two concepts common in pretraining appear in an uncommon pairing? Furthermore, current evaluations on LMMs are largely generic and do not consider the role of pretraining data (e.g., Thrush et al., 2022; Ma et al., 2022; Hsieh et al., 2023; Wang et al., 2024).

In this work, we investigate CLIP and LMM accuracies through the lens of the co-occurrence rate of concept pairs in the pretraining dataset (Figure 1). In particular, we focus on co-occurrence between words in the textual captions of CLIP pretraining examples as a proxy for visual concepts. To decorrelate the pair frequency from single-concept frequencies (i.e., the individual concepts in low frequency pairs are often themselves low frequency), we calculate pointwise mutual information (PMI) (Church & Hanks, 1990) for all concept pairs, which measures the probability of concept co-occurrence normalized by the expected probability if the concepts were independent.

We evaluate CLIP alone in a zero-shot classification setting as well as LMMs built on CLIP and find a strong correlation between concept pair PMI in pretraining data and accuracy across multiple

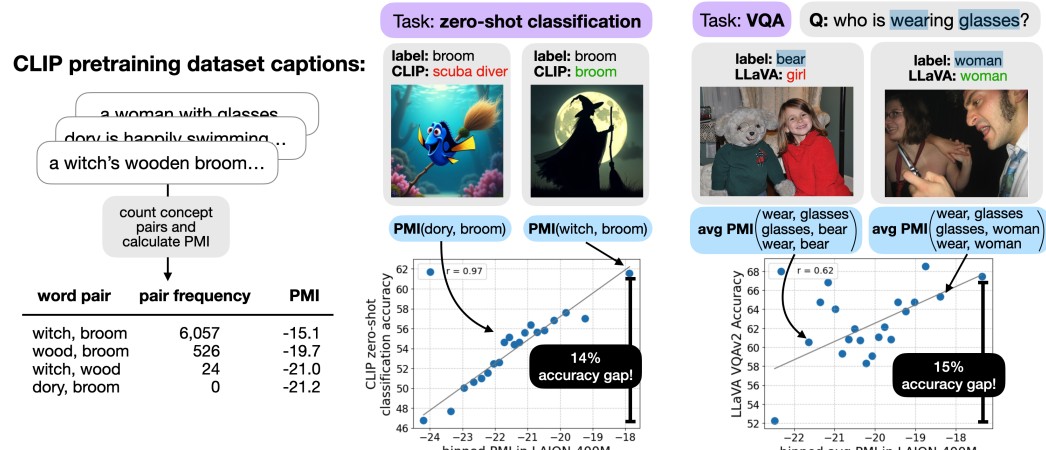

Figure 1: **Overview of our contributions. (left)** We extract concept pairs from pretraining data caption text and calculate their co-occurrence frequency and pointwise mutual information (PMI) across all captions in the dataset, including pairs that do not co-occur in the dataset at all. **(middle)** We evaluate on a dataset of real images as well as a synthetic dataset designed to contain concept combinations across a wide range of PMI (shown) and find a strong correlation between CLIP zero-shot classification accuracy and PMI (i.e., the same concept "broom" is less accurately identified in a low vs. high PMI pair). **(right)** We continue to find an accuracy vs. PMI correlation in large multimodal models built on CLIP embeddings evaluated on visual question-answering tasks. In this case, per-example PMI is averaged over all the concept pairs in the question-answer pair (shown in blue highlights).

datasets and tasks, suggesting that CLIP and LMMs have biases that depend heavily on co-occurrence statistics from pretraining rather than an understanding of the individual concepts. Specifically:

- We evaluate CLIP on a dataset of real images as well as synthetic images constructed with a variety of concept pairs, including concept pairs unseen in pretraining, and find up to a 14% absolute (30% relative) difference in zero-shot classification accuracy between images in the bottom vs. top 5% of concept pair PMI.

- With this understanding, we are able to induce an accuracy drop in ImageNet by simply pasting in a small image of an object that rarely co-occurs in pretraining with the main image object (low PMI), reducing CLIP zero-shot classification accuracy by up to 10%.

- We evaluate LLaVA, a leading LMM built on CLIP, and find that the correlation between PMI and task accuracy still holds on visual question-answering benchmarks. This result even extends to LMMs built on closed-source embedding models such as OpenAI's CLIP.

- Co-occurrence statistics in the pretraining data also lead to model biases: in particular, we find that LLaVA exhibits an output bias that is correlated with co-occurrence, tending to output "Yes" for questions with highly co-occurring concepts regardless of the true label.

- PMI-guided data curation can be an effective method for improving accuracy and robustness on rare concept combinations. We fine-tune CLIP with concept pairs covering a range of PMI values and improve classification accuracy degradation in natural images by 2.6% (28% relative).

## 2 SETUP

We introduce the models and pretraining dataset used in our analyses.

**CLIP.** Contrastive Language-Image Pretraining (CLIP, Radford et al., 2021) is a self-supervised learning method that uses natural language supervision in the form of image captions to learn downstream task-agnostic image representations. Formally, in a batch of $N$ image-text pairs $\{(x_i, t_i)\}_{i=1}^{N}$

where $x_i \in \mathcal{X}$, $t_i \sim \mathcal{T}$, CLIP simultaneously trains an image encoder $\phi : \mathcal{X} \to \mathcal{Z}_v$ and text encoder $\psi : \mathcal{T} \to \mathcal{Z}_t$, where $\mathcal{Z}_v \subset \mathbb{R}^d$, $\mathcal{Z}_t \subset \mathbb{R}^d$ denote the image and text embedding spaces, respectively. The encoders are trained to minimize the multi-class $N$-pair loss (Sohn, 2016):

$$\ell_{\text{CLIP}}(\phi, \psi) = -\frac{1}{N} \sum_i \ln \frac{e^{\phi(x_i)^\top \psi(t_i)/T}}{\sum_j e^{\phi(x_i)^\top \psi(t_j)/T}} - \frac{1}{N} \sum_j \ln \frac{e^{\phi(x_j)^\top \psi(t_j)/T}}{\sum_i e^{\phi(x_i)^\top \psi(t_j)/T}}. \tag{1}$$

For a test image $x$, we can use the learned encoders for zero-shot classification by translating a list of classification label texts $y_1, \ldots, y_k$ (where $k$ is the number of classes) into pseudo-captions $y'_1, \ldots, y'_k$, e.g., `a photo of {class name}`, and selecting the class whose pseudo-caption embedding aligns best with the image embedding: $\arg\max_i \phi(x)^\top \psi(y'_i)$.

**LMMs.** Large multimodal models (LMMs) synthesize data from multiple data modalities (e.g., image, text), typically building on top of a large language model (LLM) for natural language understanding. Many state-of-the-art open-source LMMs (e.g., LLaVA (Liu et al., 2023a; 2024a), Molmo (Deitke et al., 2024)) leverage a trained CLIP image encoder to compute visual features $\phi(x_i)$ from input image $x_i$. A vision-language connector $h : \mathcal{Z}_v \to \mathcal{Z}'_t$ is trained to map these visual features into the language model's embedding space $\mathcal{Z}'_t$. The language model and connector are then fine-tuned with conversational/question-answering data to optimize performance.

**LAION-400M.** LAION-400M (Schuhmann et al., 2021) is a dataset of 400 million image-text pairs curated from Common Crawl by filtering out pairs with CLIP embedding cosine similarity below 0.3. LAION-400M was created to emulate the closed-source WIT-400M (Radford et al., 2021) dataset used to train the original CLIP implementation.

## 3 CONCEPT PAIR EXTRACTION AND QUANTIFYING CO-OCCURRENCE

In this work, we study the impact of pretraining data on generalization to rare and unseen concept pairs in CLIP and LMMs. To this end, we define a procedure for concept extraction from large image-text datasets as well as metrics to disentangle pair co-occurrence frequency from single concept frequency.

**Concept and concept pair probability.** We define the set of concepts $\mathcal{C}$ as the set of lemmatized words extracted from a dataset of image captions $\mathcal{D}$, where a concept $c \in \mathcal{C}$ corresponds to a single lemmatized word. A concept pair is an unordered pair of concepts $(c_1, c_2)$. We define the empirical probability of single concepts as

$$p_{\mathcal{D}}(c) = \frac{1}{|\mathcal{C}|} \sum_{d \in \mathcal{D}} \mathbf{1}[c \in d] \tag{2}$$

where for simplicity we abuse the notation to define $d$ as the set of concepts derived from each caption in $\mathcal{D}$. Similarly, we define the empirical probability of a concept pair $(c_1, c_2) \in \mathcal{C} \times \mathcal{C}$ as:

$$p_{\mathcal{D}}(c_1, c_2) = \frac{1}{\binom{|\mathcal{C}|}{2}} \sum_{d \in \mathcal{D}} \mathbf{1}[c_1 \in d \ \wedge \ c_2 \in d] \tag{3}$$

We note that these definitions count the number of instances of a concept in the corpus rather than the number of documents the concept occurs in, which is a valid alternative. We provide a small ablation of this choice in Section B.2.1.

**Pointwise Mutual Information (PMI).** To decorrelate concept pair probabilities $p_{\mathcal{D}}(c_1, c_2)$ from their constituent single concept frequencies $p_{\mathcal{D}}(c_1)$, $p_{\mathcal{D}}(c_2)$, we measure the pointwise mutual information (PMI, Church & Hanks, 1990) between concept pairs:

$$\text{pmi}_{\mathcal{D}}(c_1, c_2) = \log\left(\frac{p_{\mathcal{D}}(c_1, c_2)}{p_{\mathcal{D}}(c_1) p_{\mathcal{D}}(c_2)}\right) \tag{4}$$

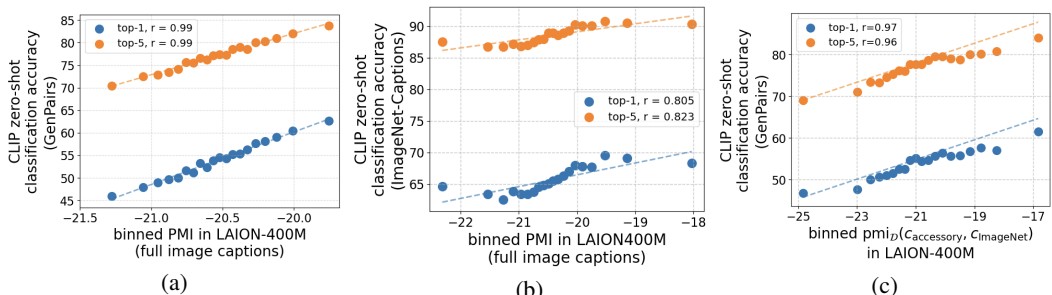

(a)        (b)        (c)

Figure 2: **Strong correlation between concept PMI in pretraining data and CLIP zero-shot classification accuracy. (a)** We evaluate LAION-400M-trained CLIP on GenPairs, where each image depicts at least one concept, $c_{\text{accessory}}$, in addition to the target ImageNet class $c_{\text{ImageNet}}$. We observe a clear correlation between average PMI over all concepts in each image caption and CLIP zero-shot top-1 and top-5 accuracies, showing that pretraining concept co-occurrence strongly influences accuracy. **(b)** We additionally evaluate on a dataset of natural images, ImageNet-Captions, and observe a similar correlation between accuracy and average PMI over the full image caption. **(c)** On GenPairs, the correlation still holds when calculated on accuracy and PMI of just the key concept pair, $(c_{\text{accessory}}, c_{\text{ImageNet}})$.

PMI measures how much more $c_1, c_2$ co-occur in $\mathcal{D}$ than we would have expected them to appear by chance. Note that while our analysis focuses on concept pairs, the PMI framework can be extended to any number of concepts $(c_1, c_2, ..., c_n)$ through specific correlation (Van de Cruys, 2011):

$$\text{si}_{\mathcal{D}}(c_1, ..., c_n) = \log\left(\frac{p_{\mathcal{D}}(c_1, ..., c_n)}{\prod_{i=1}^{n} p_{\mathcal{D}}(c_i)}\right) \tag{5}$$

**Concept Extraction and PMI Calculation in LAION-400M.** While visual concepts can be difficult to define and extract from images, we leverage the textual captions from LAION-400M as a proxy for the visual concepts present in each image. Starting with the set of LAION-400M captions, we remove stopwords and lemmatize each word. The remaining 21,718 unique words make up our concept set. To calculate PMI, we count individual frequencies as well as pair frequencies for all concepts in the set.

**Evaluation metrics.** We introduce the metrics we use to measure the relationship between PMI and task performance. On downstream datasets, we measure zero-shot accuracy for CLIP and VQA accuracy (as defined in Agrawal et al. (2015)) for LMM evaluation tasks (evaluation details in Appendix A.6). We define *accuracy gap* as the accuracy difference between inputs in the top and bottom 5% of PMI values, representing the absolute accuracy degradation due to low PMI inputs. We also report Pearson $r$ correlation coefficients to quantify correlation strength and direction.

## 4   CLIP ZERO-SHOT CLASSIFICATION ACCURACY CORRELATES WITH CONCEPT PMI

We investigate the relationship between PMI and CLIP zero-shot accuracy through a dataset of synthetic images with a controlled variety of concept pairs, which we call *GenPairs*, as well as a dataset of real images with associated captions, ImageNet-Captions (Fang et al., 2022).

**Task.** We test CLIP in the zero-shot classification setting on two datasets: GenPairs and ImageNet-Captions. For GenPairs, we design input images for zero-shot classification that each feature at least two concepts, one of which is an ImageNet class. ImageNet-Captions is a subset of 463,622 images from the ImageNet training dataset augmented with the text data associated with the original Flickr sources.

**GenPairs concept pairs.** For the GenPairs evaluation, we generate synthetic data using concept pairs that span the range of PMI in LAION-400M. We first identify concept pairs (from the set of concepts extracted from LAION-400M) where only one of the two concepts is an ImageNet class.

To do so, we create a set of ImageNet class *categories*, defined by the last word of each class name (e.g., `king charles spaniel → spaniel`). We select pairs where one of the two concepts matches an ImageNet category name and the other does not, including concept pairs that do not co-occur at all in LAION-400M as long as each individual concept is present. Let such a pair be denoted $(c_{\text{accessory}}, c_{\text{ImageNet}})$, where $c_{\text{ImageNet}}$ is the ImageNet class word. Finally, we filter the set of $c_{\text{accessory}}$ to those that can be visualized in an image. Further details can be found in Appendix A.2.

**GenPairs dataset construction.** We construct GenPairs by generating a synthetic image for each concept pair extracted from LAION-400M where one of the two concepts is an ImageNet class, and define that ImageNet class as the ground truth label. We subsample the set of concept pairs to obtain 200,000 pairs across the range of concept PMI. We use Llama 3.1 8B Instruct (Grattafiori et al., 2024) to generate a realistic caption for an image that features each concept pair, and use these captions to prompt Flux.1-dev (Black Forest Labs, 2024) to generate images (details in Appendix A.2). We empirically find that Flux.1-dev produces realistic images even for low PMI pairs (see Figure 6 for examples from GenPairs).

**CLIP zero-shot classification accuracy correlates strongly with PMI of image caption concepts.** We evaluate a CLIP ViT-B/32 pretrained on LAION-400M on zero-shot classification with the images in GenPairs and ImageNet-Captions (following the protocol defined in Radford et al. (2021)). For each dataset, we calculate the average PMI over all valid concept pairs for each example, bin them into 20 equally sized bins, and show the correlation with classification accuracy in Figure 2a, 2b. We observe an $r = 0.99$ correlation between PMI and CLIP zero-shot top-1 classification accuracy and an accuracy gap of 18% on GenPairs (Figure 2a), and $r = 0.81$ and an accuracy gap of 5% on ImageNet-Captions (Figure 2b). Our results indicate that instead of generalizing to rarely seen concept combinations, CLIP's classification accuracy correlates predictably with the pretraining co-occurrence rate of the depicted concepts in the image.

**CLIP zero-shot classification accuracy correlates strongly with PMI of key concept pair.** Since the captions in GenPairs were explicitly generated to contain the concept pair $(c_{\text{accessory}}, c_{\text{ImageNet}})$, we term this the "key concept pair". This construction allows us to understand how the additional concepts in the caption impact the PMI-accuracy correlation. To this end, we analyze the correlation between zero-shot classification accuracy and the single PMI value for the key concept pair, $\text{pmi}_{\mathcal{D}}(c_{\text{accessory}}, c_{\text{ImageNet}})$ (Figure 2c). We observe a $r = 0.97$ correlation and an accuracy gap of 14%. This suggests that the PMI of the key concept pair alone is predictive of CLIP classification accuracy.

## 5 CLIP'S ZERO-SHOT CLASSIFICATION ACCURACY CORRELATES WITH PMI IN EDITED NATURAL IMAGES

We observed in the previous section that while images often contain many concepts, a single key concept pair can be sufficient for analyzing the relationship between PMI and CLIP's zero-shot accuracy. In this section, we use this insight to construct edits to ImageNet images that introduce a particular concept pair to the image, affecting accuracy.

**Task.** We test CLIP in the zero-shot classification setting on images altered to contain a specific concept pair. In order to control the concept combinations in an image, we edit ImageNet validation set images by pasting a small image of an accessory concept. We then evaluate how CLIP's accuracy on edited images correlates with the PMI of the concept pair of the label and the pasted concept.

**Concept pairs.** We use the set of concept pairs defined in Section 4 where one of the two concepts is an ImageNet class. Each pair can be denoted $(c_{\text{accessory}}, c_{\text{ImageNet}})$, where $c_{\text{ImageNet}}$ is the class of an ImageNet image.

**ImageNet-Paste dataset construction.** We construct our evaluation dataset, which we call *ImageNet-Paste*, in two stages: first, we generate images of the set of accessory concepts using Flux.1-dev (details in Appendix A.3). For each ImageNet class $c_{\text{ImageNet}}$, we sample a set of accessory concepts $c_{\text{accessory}}$ across a range of PMIs $\text{pmi}_{\mathcal{D}}(c_{\text{accessory}}, c_{\text{ImageNet}})$. We scale the $c_{\text{accessory}}$ image to

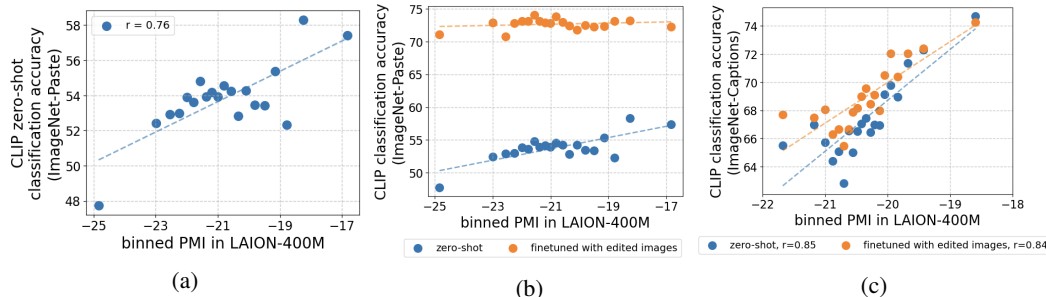

Figure 3: **(a)** Editing real (ImageNet validation) images by pasting an image of a concept with known PMI with the ImageNet class induces a correlation between zero-shot accuracy and PMI of the pasted concept and target class. In particular, pasting an image with a low PMI with the target class results in a 10% accuracy drop. **(b)** Fine-tuning CLIP with edited images improves the overall accuracy and removes the correlation between PMI and accuracy on a held-out set of ImageNet-Paste. **(c)** Accuracy improvements from fine-tuning can transfer to other datasets: evaluating CLIP fine-tuned on edited images improves accuracy across the PMI spectrum on a subset of ImageNet-Captions that shares concept pairs with the fine-tuning dataset.

be at most 10% of the original image size, and paste it onto a randomly selected location (see Figure 7 for examples from ImageNet-Paste).

**Strong correlation between PMI of edited ImageNet images and CLIP zero-shot classification accuracy.** Figure 3a shows the correlation between CLIP top-1 zero-shot classification accuracy and $\text{pmi}_{\mathcal{D}}(c_{\text{accessory}}, c_{\text{ImageNet}})$ in ImageNet-Paste ($r = 0.75$) with an accuracy gap of 10%. We note that the correlation we find is independent of the documented failures of CLIP models on classes that are poorly represented in pretraining, since the PMI metric is normalized by the individual concept frequencies. Our concept-pair framework reveals a clear vulnerability in CLIP: even simple interventions, like pasting a concept with low PMI relative to the target class, can significantly degrade performance.

**Fine-tuning CLIP with edited images substantially reduces the correlation when tested on edited images.** We assess the impact of PMI-based image editing as an augmentation strategy for improved generalization to low PMI inputs. We follow the procedure for generating image edits described above but implement it as an on-the-fly augmentation applied during a fine-tuning step to optimize CLIP embeddings to the ImageNet classification task. Specifically, we perform end-to-end fine-tuning on the CLIP model for the ImageNet classification task with a linear projection layer (see Appendix A.4 for implementation details). We evaluate CLIP fine-tuned with the augmentation on a held-out set of edited images and show our findings in Figure 3b. After fine-tuning with augmentation, the accuracy gap is reduced to 1% compared to 10% zero-shot.

**Robustness gains from fine-tuning with edited images can transfer to other datasets.** To determine if fine-tuning with the image editing augmentation procedure can be a general strategy for improving accuracy on low PMI pairs, we evaluate fine-tuned CLIP with the subset of ImageNet-Captions (45k examples) that shares concept pairs with the fine-tuning dataset (Figure 3c). On this subset, zero-shot accuracy decreases from 74.7% to 65.5% between highest and lowest PMI bins, creating a 9.2% accuracy gap. We find that the accuracy drop after fine-tuning with edited images is 74.3% → 67.7%, making the accuracy gap 6.6% (-2.6% compared to no fine-tuning). We additionally find that evaluating fine-tuned CLIP on a subset of GenPairs that shares concept pairs with the fine-tuning dataset (10k examples) reduces accuracy gap from 7% → 2%. These results suggest that the benefit from fine-tuning can transfer to other natural and synthetic datasets as long as the fine-tuning dataset is sufficiently similar in concept pair coverage.

## 6 LMM PERFORMANCE CORRELATES WITH CONCEPT PMI

In this section, we extend our analysis from CLIP to large multimodal models (LMMs) that incorporate CLIP in their architecture. We find that CLIP's failures to generalize to low PMI concept pairs affect downstream LMMs built on CLIP embeddings.

**Task.** We evaluate LMMs on the visual question-answering (VQA) task, which tests multimodal models' ability to understand visual inputs through open-ended natural language questions about images. A VQA input example consists of an image and a natural language question about the image, and a set of possible ground truth answers produced by human annotators. We identify concepts and calculate the PMI of each input VQA example by analyzing the question and answer text, and assess the LMM's ability to respond correctly to the question as a function of PMI.

**Model.** For this analysis, we use two variants of the LLaVA-1.5-7B model (Liu et al., 2023a; 2024a), a leading LMM built on CLIP image embeddings. The publicly available LLaVA-1.5-7B uses CLIP embeddings from OpenAI-trained CLIP ViT-L/14, which we denote LLaVA-1.5-OpenAI. However, the OpenAI pretraining data is not publicly available. In order to draw a direct comparison to CLIP pretraining data, we train our own version of LLaVA-1.5-7B that uses CLIP embeddings from LAION-400M-trained CLIP ViT-L/14, which we denote LLaVA-1.5-LAION. We follow the visual instruction tuning procedure outlined in Liu et al. (2024a) to finetune a LLaVA-1.5 model with LAION-400M-trained CLIP as the vision backbone (details in Appendix A.5).

**Datasets.** We evaluate on two standard visual question-answering benchmarks, VQAv2 (Goyal et al., 2017) and TextVQA (Singh et al., 2019). VQAv2 is an open-ended VQA benchmark designed to test image understanding by targeting skills including object recognition, object counting, and relative locations. VQAv2 includes a mix of yes/no and open-ended (not yes/no) questions, which we evaluate on separately. TextVQA specifically focuses on question-answering with an optical character recognition (OCR) component. We test on the validation split of VQAv2 (since the test split ground truth answers are not publicly available) and the test split of TextVQA. We quantify performance using VQA accuracy as defined by the benchmarks.

**Concept pairs.** We adapt our PMI framework to the VQA setting by extracting concepts from the text of both the question and ground truth answer, as they both can contain information about the image (e.g., `Q: who is wearing glasses? A: woman` → {wear, glasses, woman}). We calculate PMI values for all concept pairs in each VQA example, then take the average for the final example-level PMI value.

**Strong correlation between PMI and LMM VQA accuracy.** We evaluate LLaVA-1.5-LAION on TextVQA and VQAv2 and find a clear correlation with PMI. In particular:

- **TextVQA:** We find a 15% accuracy gap and a Pearson correlation coefficient of $r = 0.70$ (Figure 4, top left).
- **Open-ended VQAv2:** We observe a 15% accuracy gap on open-ended VQAv2 examples (questions that require more than a binary yes/no response) and a correlation coefficient of $r = 0.62$ (Figure 4, bottom left).
- **Yes/No VQAv2:** We find a strong correlation for yes/no questions in VQAv2, with $r = 0.80$ and a 4% accuracy gap (Figure 5a).

We note that despite incorporating an LLM and performing visual instruction tuning steps that expose the full multimodal model to additional data, biases in the visual encoder still affect the performance of the downstream LMM.

**Closed-source models exhibit an almost identical correlation between PMI and VQA accuracy.** We additionally test LLaVA-1.5-OpenAI on both benchmarks. Despite the fact that OpenAI CLIP is trained on a closed-source dataset with few known properties, we observe that the correlation plots look almost identical to those of LLaVA-1.5-LAION other than an overall shift in accuracy. We find a 10% accuracy gap on TextVQA ($r = 0.76$) and 13% on VQAv2 ($r = 0.73$) (Figure 4, right). This

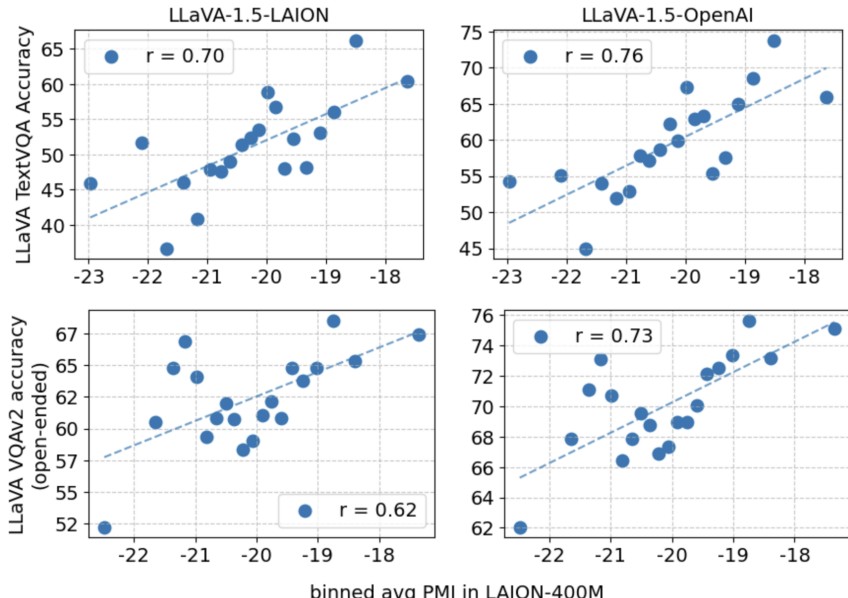

Figure 4: **Strong correlation between PMI in LAION-400M and LLaVA accuracy on VQA tasks.**
We observe a strong correlation between LAION-400M LLaVA performance on both TextVQA (**top row**) and VQAv2 (**bottom row**), where PMI for each input example is averaged across all concept pairs in the question and answer text. For VQAv2, we report performance on open-ended questions (all questions that require more than a 'yes/no' response). LLaVA built on OpenAI CLIP (**right column**) also exhibits an almost identical correlation when using PMIs calculated with LAION-400M, despite OpenAI CLIP not being pretrained on LAION-400M.

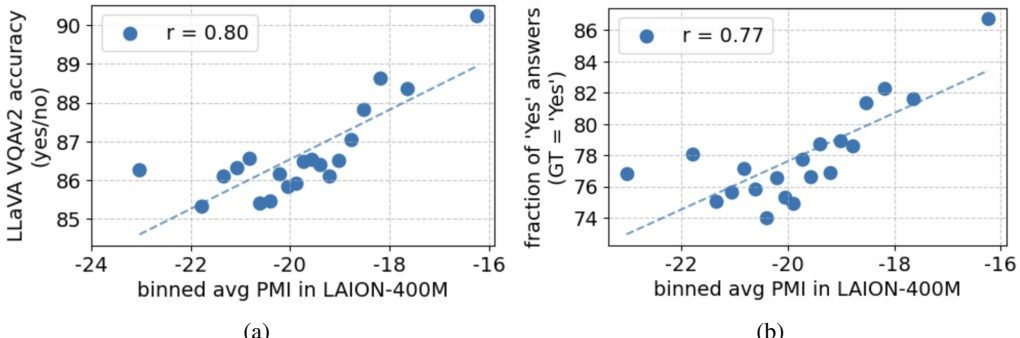

(a)                                   (b)

Figure 5: **(a) Strong correlation between PMI in LAION-400M and LLaVA accuracy on VQAv2 yes/no questions. (b) LMMs respond 'yes' more often for higher average PMI inputs regardless of the true answer.** For VQAv2 questions with ground truth answer 'yes', we find the rate at which LAION-400M LLaVA correctly responds 'yes' is highly correlated with the average PMI of concept pairs in the input question.

suggests a shared long-tailed distribution of concept pairs in web-crawled datasets, and points to the possibility of performing data-centric analyses of closed-source model accuracy with open-source datasets.

**LMMs bias toward responding 'yes' with increasing PMI of concepts in the question.** We additionally find that, for questions in the VQAv2 dataset with ground truth answer 'yes', LLaVA-1.5-LAION's likelihood of correctly responding 'yes' is positively correlated ($r = 0.77$) with the PMI of the concepts in the question (Figure 5b). Here, we recalculate PMI with the concepts from the question only, not the answer (i.e., remove 'yes' and 'no' from the concept pairs), and find that the

presence of high PMI concept pairs in the question is more likely to elicit a 'yes' response. Intuitively, this means that if concepts in the question co-occur often in the CLIP pretraining data, then the model is likely to respond 'yes' regardless of the actual content in the image. We note that this is an opposing effect to the correlation between PMI and accuracy described throughout this paper, as this bias worsens with higher PMI. In this case, however, the overall accuracy improves despite this bias.

# 7 RELATED WORK AND DISCUSSION

**Spurious correlations.** Machine learning models tend to learn spurious correlations present in the training dataset (Beery et al., 2018; Zech et al., 2018; Sagawa et al., 2020; Geirhos et al., 2020); however, these works primarily investigated supervised training settings, where spurious features connect the input with the prediction target. Recent work (Wang et al., 2024) demonstrates that even web-scale pretrained models evaluated under zero-shot conditions are not immune to basic spurious correlations, such as wildlife in likely vs. unlikely environments, but the connection to pretraining data was not explored. Our work goes a step further to study the relationship between frequency of concept co-occurrence in pretraining to downstream accuracy. Data reweighting interventions such as Group DRO (Sagawa et al., 2020) required group labels, while our PMI metric could enable new data-centric algorithms for robustness to distribution shifts for multimodal data containing text.

**CLIP robustness.** A key signature of CLIP is its strong zero-shot accuracy across a range of historically challenging datasets (Radford et al., 2021), indicating that a large, diverse pretraining dataset may be sufficient for robust generalization (Fang et al., 2022; Mayilvahanan et al., 2024; Xue et al., 2023). However, recent work has shown that CLIP's downstream accuracy is notably worse on examples involving concepts that are poorly represented in the training data (Udandarao et al., 2024; Parashar et al., 2024), and a few evaluate CLIP's performance on concept combinations unseen during pretraining (Abbasi et al., 2024; Wiedemer et al., 2025). However, these works both lack evaluation on CLIP-based LMMs and fail to take into account the frequency of the concept *combination*, only the frequencies of each individual concept in pretraining. We introduce the PMI metric, which allows us to take into account the pretraining frequency of the combination of concepts as well as the frequencies of the individual concepts. We investigate the full range of unseen/rare to common concept combinations in pretraining and find a strong correlation with CLIP and CLIP-based LMM accuracy, indicating that accuracy degradation occurs even on common concepts when they appear in uncommon pairings. Our findings highlight the need for algorithms and architectures that improve generalization in multimodal models without scaling the training data combinatorially.

**LMMs and evaluation.** Modern LMMs are primarily built by combining embeddings from a frozen visual encoder, most commonly CLIP, with a large language model (Li et al., 2023a; Liu et al., 2023a; Awadalla et al., 2023; Deitke et al., 2024; Liu et al., 2024a; Tong et al., 2024a). As such, failures of the visual encoder can have a direct impact on the efficacy of the downstream LMM (Tong et al., 2024b). Existing evaluations of LMMs test for a wide variety of capabilities (Singh et al., 2019; Goyal et al., 2017; Hua et al., 2024; Tong et al., 2024a; Ma et al., 2022; Hsieh et al., 2023; Thrush et al., 2022; Liu et al., 2024b), but the connection between task performance and the pretraining data distribution of the visual encoder has not been explored. We fill this important gap by demonstrating that the relationship between concept PMI in pretraining and CLIP performance on those concepts extends to CLIP-based LMMs, underscoring the importance of a robust visual encoder.

**Effect of model scale.** Motivated by prior work showing the link between model scale and improved generalization in downstream tasks (e.g., Liu et al., 2023b; Redhardt et al., 2025), we studied the effect of scaling up CLIP model size on the observed PMI-accuracy correlation on CLIP zero-shot accuracy as well as in the context of a LMM. We find limited improvement in the accuracy gap: $14.8\% \rightarrow 13.4\%$ between our smallest (ViT-B/32) and largest (EVA01-g/14) models, despite a nearly 40x increase in FLOPs between the two models. We also train LLaVA-1.5-7B models with varying CLIP model size and find that the relationship between accuracy gap and model size varies between tasks and, in the best case (open-ended questions from VQAv2), the accuracy gap decreased from 15.8% to 14% between the smallest and largest models. We conclude that simply scaling CLIP is not sufficient to consistently improve accuracy on rare concept pairs. Additional details on our scaling experiments can be found in Appendix B.1.

## 8 CONCLUSION

Our study reveals that CLIP and LMMs built on CLIP are highly sensitive to the co-occurrence statistics of concept pairs in their pretraining data. This leads to a strong correlation between PMI of inputs and task accuracy as well as sizable observed accuracy gaps between high and low PMI concept pairs across multiple datasets and tasks, showing that these models struggle to disentangle individual concepts and generalize to new combinations. While interventions like scaling model size did not produce consistent gains, we find that fine-tuning with a broad range of concept pair PMI is a promising avenue for further investigation. We conclude that closing this gap will require new methods that promote robust generalization without relying on combinatorially large datasets.

### REPRODUCIBILITY

To encourage reproducibility, we submit all code as supplementary material. Additionally, we describe the details of our dataset generating pipeline (for GenPairs and ImageNet-Paste) as well as our model training/fine-tuning hyperparameters in Appendix A. Upon publication, we will release the code and datasets to the community.

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

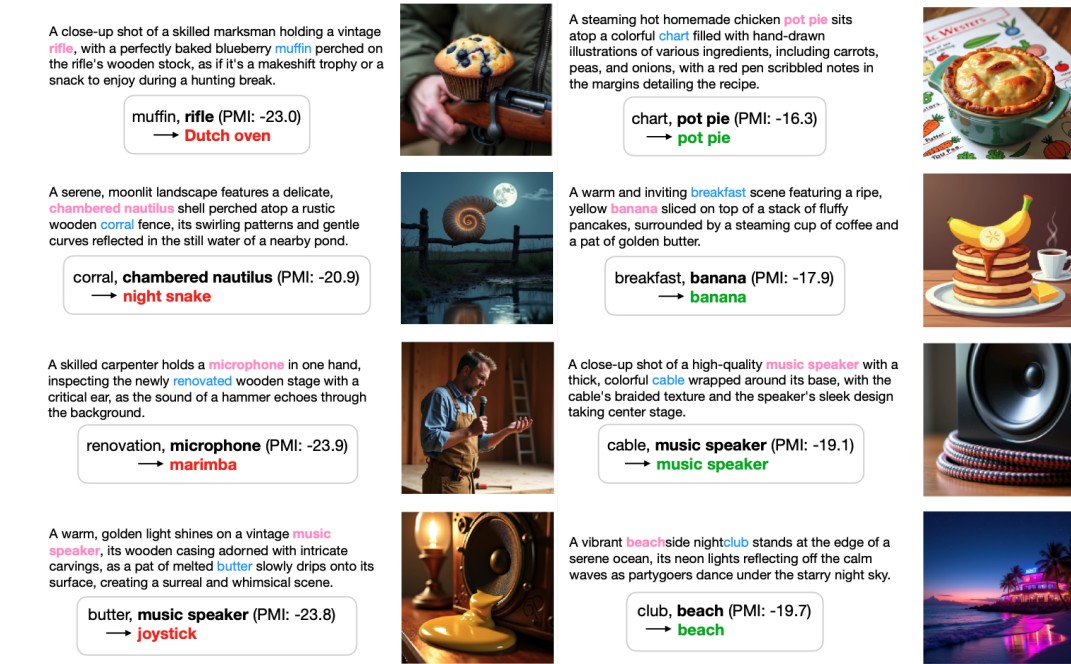

Figure 6: **Examples from GenPairs** (**left**: low PMI, **right**: high PMI). We use Meta's Llama 3.1 8B Instruct to generate captions for images incorporating the concept pairs ($c_{\text{accessory}}$ in pink, $c_{\text{ImageNet}}$ in bold/blue). We prompt Flux.1-dev with the captions to produce the images shown. Finally, CLIP's zero-shot prediction on each image is shown (red: incorrect, green: correct).

## A IMPLEMENTATION DETAILS

### A.1 ADDITIONAL DETAILS ON CONCEPT EXTRACTION AND PMI CALCULATION

We use the `nltk` package to clean, lemmatize, and perform part-of-speech tagging for the LAION-400M captions. To treat pairs with $p_{\mathcal{D}}(c_1, c_2) = 0$ or $p_{\mathcal{D}}(c) = 0$, we calculate all PMI frequency ratios with Laplace smoothing with a smoothing factor of $\alpha = 1$ for $p_{\mathcal{D}}(c_1, c_2)$ and $\alpha = 1e4$ for $p_{\mathcal{D}}(c)$.

### A.2 SYNTHETIC DATA GENERATION

After obtaining the set of $(c_{\text{accessory}}, c_{\text{ImageNet}})$ concept pairs, we want to retain a set of $c_{\text{accessory}}$ that are visualizable English words. To do this, we first perform basic filtering: we remove $c_{\text{accessory}}$ with numeric digits or that are not in a WordNet synset (as a proxy for non-English words or non-words), then perform POS tagging and keep only nouns and adjectives that are not the words `photo` or `image`. Finally, we prompt Llama 3.1 8B Instruct to distinguish $c_{\text{accessory}}$ that are "visualizable" in order to filter out words like "new" or "success" that are difficult to incorporate into an image. We reproduce the prompt in Block 1.

After filtering, we generate a one-sentence image caption for each concept pair using Llama 3.1 8B Instruct and the prompt in Block 2. We then prompt the text-to-image model Flux.1-dev with the captions produced in the previous step. The hyperparameters we used for Llama and Flux.1-dev are detailed in Tables 1 and 2. We empirically find that these hyperparameters produce the most realistic captions and images for our purposes. We use HuggingFace implementations of both models.

We use the OpenCLIP implementation of all models (Cherti et al., 2023) and note that EVA01-g/14 is from the EVA-CLIP family of models (Sun et al., 2023).

```
You will be provided with some examples of questions and answers
    determining whether a word is easily visualizable, followed by a
    question for you to solve. An easily visualizable word is a concrete
```

```
    thing or adjective that describes the subject of an image. Abstract
    concepts that can be represented by concrete objects/images are NOT
    easily visualizable. When in doubt, answer no. Please think aloud
    step-by-step and conclude your answer with the phrase "The answer is
    X.". You must use exactly this phrase, otherwise we will be unable to
     use your answer.

## Examples

Q: Is temperament easily visualizable?
A: Let's think step by step. Temperament is a property of a person/animal
    , so the subject of the image would be that person/animal and not "
    temperament". The answer is no.

Q: Is sb easily visualizable?
A: Let's think step by step. Sb is not a word and is thus not
    visualizable. The answer is no.

Q: Is fertilizer easily visualizable?
A: Let's think step by step. Fertilizer is a concrete object and can be
    visualized by, e.g., a bag of fertilizer. The answer is yes.

Q: Is impressionism easily visualizable?
A: Let's think step by step. Impressionism is an art style so images can
    be rendered in an impressionist style. The answer is yes.

Q: Is browsing easily visualizable?
A: Let's think step by step. Browsing is an action, and actions are not
    directly visualizable in a static image. The answer is no.

Q: Is success easily visualizable?
A: Let's think step by step. Success is an abstract concept. It could be
    represented by a trophy or other concrete object, but then that
    object would be the subject of the image, so it is not directly
    visualizable. The answer is no.

Q: Is helen easily visualizable?
A: Let's think step by step. Helen is a proper noun, likely referring to
    a person named Helen, but this would be impossible to know without a
    text description. Helen is thus not visualizable. The answer is no.

## Your Question
Q: Is {c} easily visualizable?
A: Let's think step by step.
```

Listing 1: Prompt used for $c_{\text{accessory}}$ "visualizability" filtering with Llama. `{c}` is replaced with the concept word.

```
Please write a single sentence that could describe an image that contains
     the words '{c1}' and '{c2}'. Make sure both {c1} and {c2} are the
    focus of the image.
```

Listing 2: Prompt used for image caption generation with Llama. `{c1}` and `{c2}` are replaced with the concepts in the concept pair.

### A.3 NATURAL IMAGE EDITING

We generate an image of each $c_{\text{accessory}}$ by prompting Flux.1-dev with the simple phrase "a {c_accessory} in the center of a white background". As these accessory images will be pasted on top of ImageNet images, we replace the pasted images' white background with transparent pixels to emulate the concept occurring in the image "naturally". To do so, we generate an object mask with the Segment Anything (Kirillov et al., 2023) object segmentation model and assign

| parameter | value |
|---|---|
| temperature | 0.1 |
| minp | 0.05 |
| max new tokens | 50 |

Table 1: Llama hyperparameters for visualizability filtering and caption generation.

| parameter | value |
|---|---|
| output size (px) | $512 \times 512$ |
| guidance scale | 5.0 |
| inference steps | 28 |

Table 2: Flux.1-dev hyperparameters for generating the images for Section 4 as well as the pasted images for Section 5.

masked background pixels to fully transparent using the RGBA format. All image manipulations were done with the `PIL` Python package. Examples of edited images are shown in Figure 7.

### A.4 CLIP FINE-TUNING

We fine-tune CLIP for the ImageNet classification task starting with a linear layer initialized with CLIP's zero-shot ImageNet classification weights. We follow the WiSE-FT fine-tuning recipe (Wortsman et al., 2021) to fine-tune end-to-end with learning rate 3e-5 with 500 steps of linear warmup, weight decay 0.1, batch size 512, and train for 10 epochs. We train with the ImageNet training split with the image editing augmentation described in Section 5.

### A.5 LLAVA EXPERIMENTS

**LLaVA fine-tuning.** We fine-tune our own LLaVA-1.5-7B with LAION-400M CLIP ViT-L/14. We follow the LLaVA-1.5 visual instruction tuning recipe and first pretrain the vision-language connector with CLIP and LLM both frozen. We keep the same hyperparameters as they recommend but empirically find that a lower maximum learning rate of 1e-4 is more effective. After the pretraining step, we fine-tune both the connector and the LLM using the published LLaVA-1.5 visual instruction tuning dataset with the suggested hyperparameters.

**Processing the VQA datasets.** As TextVQA and VQAv2 answers come from real human responses with some variance, we define the "ground truth answer" as the mode of the collected human responses. We tokenize, lemmatize, and remove stopwords (we do not remove 'yes' and 'no' since some examples are yes/no questions) from each question-answer pair in the VQA datasets to obtain the set of concepts and concept pairs for each example.

### A.6 EVALUATION DETAILS

We follow the OpenCLIP implementation (Cherti et al., 2023; Ilharco et al., 2021) of the original CLIP (Radford et al., 2021) work's zero-shot classification recipe for all zero-shot CLIP evaluations on GenPairs and ImageNet-Paste. We follow the directions in the official LLaVA repository[1] to evaluate LLaVA-1.5-LAION and LLaVA-1.5-OpenAI on TextVQA and VQAv2.

---

[1]https://github.com/haotian-liu/LLaVA/blob/main/docs/Evaluation.md

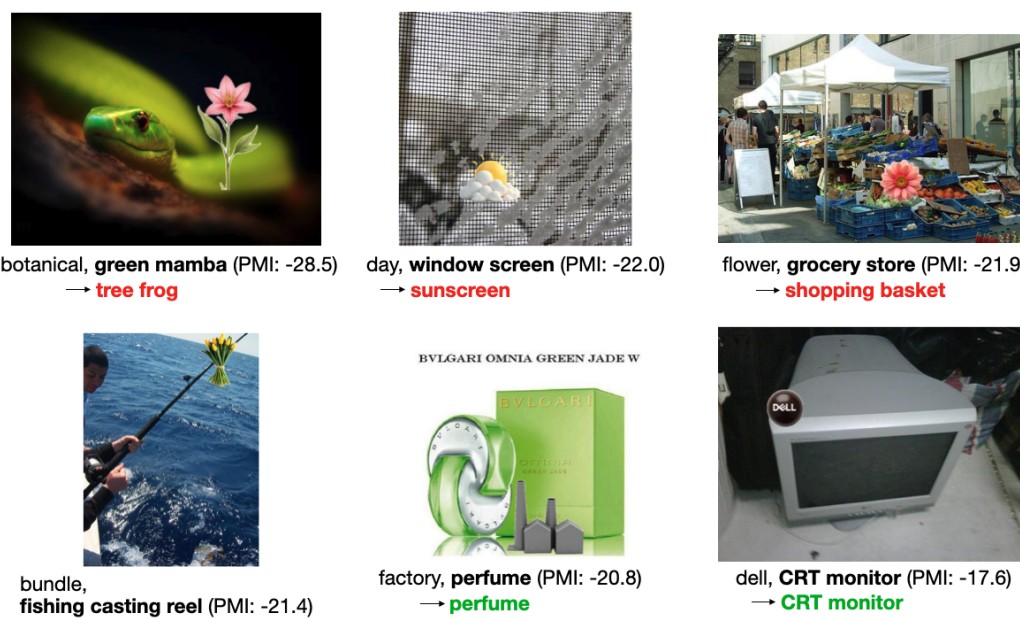

Figure 7: **Examples from our edited natural images dataset described in Section 5.** We prompt Flux.1-dev to generate images of a set of $c_{\text{accessory}}$ accessory concepts, then paste onto an ImageNet validation set image of class $c_{\text{ImageNet}}$. The concept pair is shown under each image, with $c_{\text{ImageNet}}$ shown in bold, as well as CLIP's zero-shot prediction on each edited image (red: incorrect, green: correct).

## B  ADDITIONAL EXPERIMENTS

### B.1  SCALING EXPERIMENTS

#### B.1.1  CLIP MODEL SCALING OFFERS LIMITED ROBUSTNESS GAINS

Constructing a dataset with a balanced distribution over all concept combinations may reduce PMI-driven bias, but this approach becomes intractable with the number of possible combinations. Instead, we investigate the impact of scaling the *model* rather than the *dataset*. In addition to the ViT-B/32 baseline that is used for all CLIP experiments, we test 3 larger models all pretrained with LAION-400M (in increasing order of size: ViT-B/16+ 240, ViT-L/14, and EVA01-g/14) on GenPairs. While the correlation persists across model scale (Figure 8a), Figure 8b shows that the accuracy gap decreases slightly from 14.8% in the smallest model, ViT-B/32, to 13.4% in the largest model, EVA01-g/14 (Sun et al., 2023).

#### B.1.2  SCALING CLIP IN THE LMM CONTEXT

We extend our finding regarding the impact of CLIP model scale on robustness to LMMs. Specifically, we train LLaVA-1.5-7B models with the 4 LAION-400M-pretrained CLIP backbones from Figure 8: ViT-B/32, ViT-B/16+ 240, ViT-L/14, and EVA01-g/14 (in increasing order of size). We note that the default CLIP model size for LLaVA-1.5-7B is ViT-L/14, which was used for all other LLaVA experiments in this section. We observe that correlation between PMI and VQA accuracy persists across scales (Figure 9, top row) and the relationship between accuracy gap and model size varies between tasks (Figure 9, bottom row); however, the largest model (EVA01-g/14) consistently produces a smaller accuracy gap compared to the smallest (ViT-B/32) ($15.8 \rightarrow 14.0$ for VQAv2 (open-ended), $3.8 \rightarrow 3.1$ for VQAv2 (yes/no), $16.1 \rightarrow 15.3$ for TextVQA). These results suggest that, in the context of CLIP-based LMMs, scaling CLIP alone may not consistently produce robustness gains.

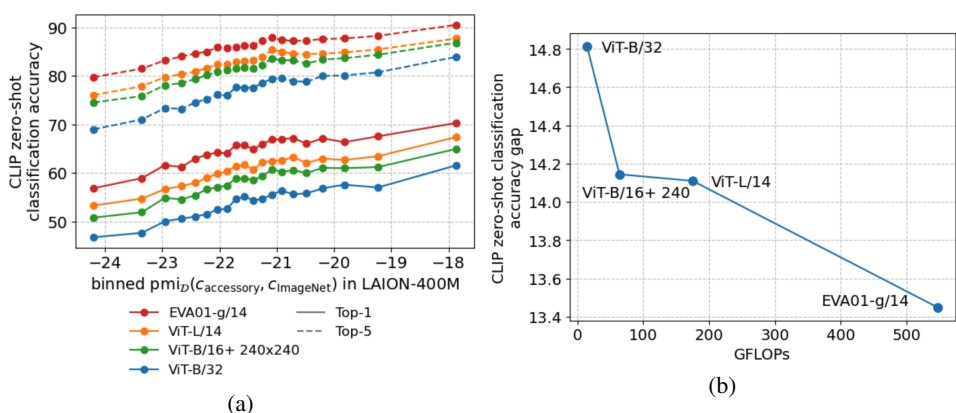

(a)

(b)

Figure 8: **Accuracy gap improves slightly with model scale. (a)** In addition to ViT-B/32, we test 3 additional CLIP architectures pretrained with LAION-400M on GenPairs. **(b)** Accuracy gap on zero-shot classification decreases slightly with model scale.

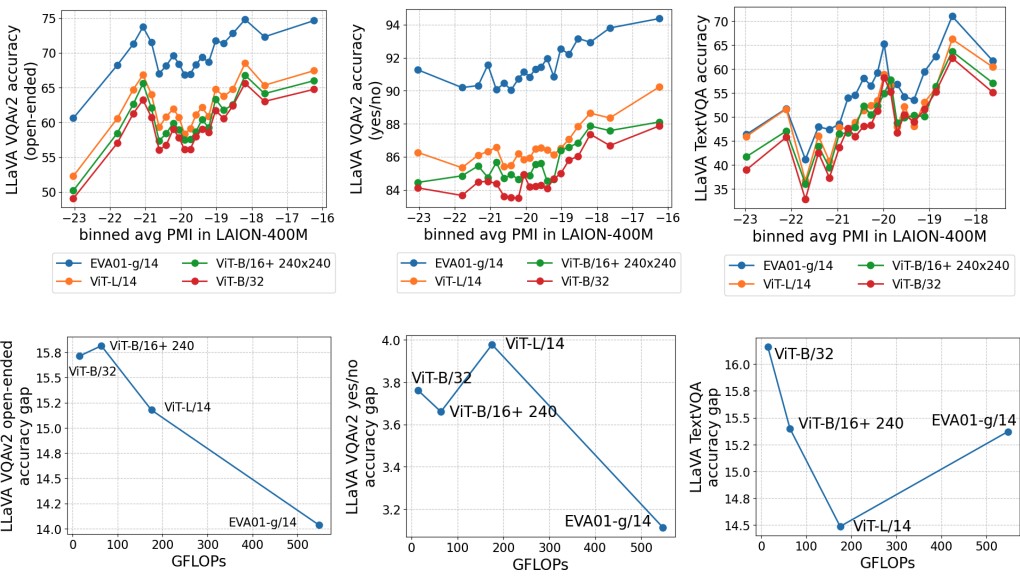

Figure 9: **CLIP model scale does not consistently improve generalization to low PMI inputs in LMMs. (top row)** In addition to the default CLIP ViT-L/14, we train LLaVA-1.5-7B models based on 3 additional CLIP architectures and test them on VQAv2 and TextVQA. **(bottom row)** CLIP model scale is not consistently predictive of accuracy gap across tasks.

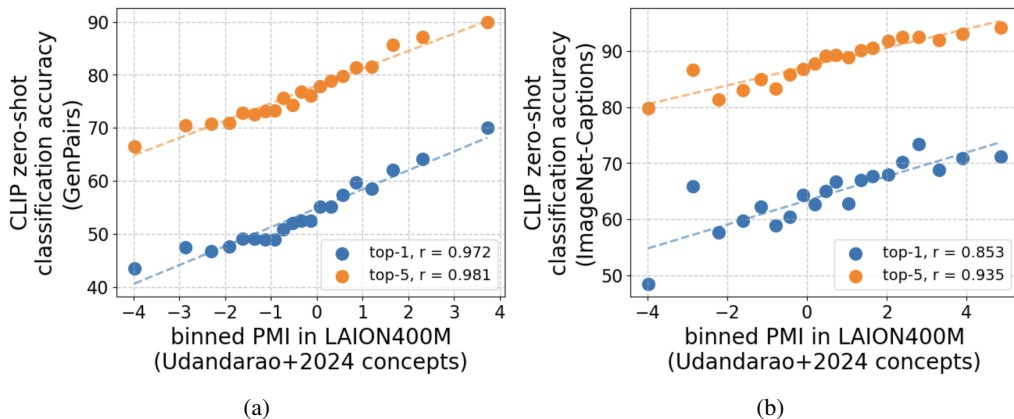

(a)  (b)

Figure 10: Correlation between zero-shot classification accuracy and PMI remains strong with PMIs calculated using alternative concept extraction pipeline from Udandarao et al. (2024).

## B.2 ABLATIONS

### B.2.1 CONCEPT EXTRACTION PIPELINE

Understanding and quantifying the distribution of concepts in the pretraining and evaluation datasets are essential to all the results presented in this work. As recent works have introduced alternative concept extraction pipelines from vision-language datasets, we ablate our concept extraction pipeline to understand the robustness of our results. We adopt the pipeline presented in Udandarao et al. (2024), which notably differs from ours in that they additionally perform concept extraction from the images themselves, rather than exclusively relying on the text captions.

As Udandarao et al. (2024) have published their single concept frequencies for LAION-400M, we simply adopt these and only implement their concept extraction pipeline on our evaluation datasets, GenPairs and ImageNet-Captions. Following the work, we define the concepts present in an (image, text) pair as the intersection of concepts detected in the image and concepts detected in the text caption, and add the ImageNet class if it was not successfully detected. We calculate PMI by slightly altering Equations 2 and 3 to reflect the document-based counting used in Udandarao et al. (2024) by replacing both normalizing constants $\frac{1}{|\mathcal{C}|}$ and $\frac{1}{\binom{|\mathcal{C}|}{2}}$ with $\frac{1}{|\mathcal{D}|}$, where $|\mathcal{D}|$ represents the total number of documents in the dataset.

Our results in Figure 10 show that using concepts extracted from this pipeline actually bolsters the original correlation we observed (Figure 2): Pearson $r$ increased for almost all correlations relative to our original results. For GenPairs (Figure 10a), Pearson $r$ correlation between PMI and top-1 accuracy is unchanged from our original results at $r = 0.97$, while $r = 0.96 \rightarrow 0.98$ for top-5 accuracy. For ImageNet-Captions (Figure 10b), we find $r = 0.81 \rightarrow 0.85$ for top-1 accuracy, and $r = 0.82 \rightarrow 0.94$ for top-5 accuracy.

### B.2.2 IMPACT OF VISUAL INPUT ON LMM 'YES' BIAS

To better understand whether LMMs' bias toward responding 'yes' to questions with higher concept PMI is due to biases in the visual or language modeling component, we ablate the presence of visual input to the LMM to isolate the LLM. Specifically, we perform the same evaluation on yes/no questions in VQAv2 described in Section 6 but replace all images with random noise. We show the result in Figure 11, which is directly analogous to Figure 5 except image inputs are replaced with noise. We find that the correlation strength decreased dramatically ($r = 0.77 \rightarrow 0.37$), indicating that the bias from the LLM alone is not sufficient to explain the much stronger bias observed with the visual input.

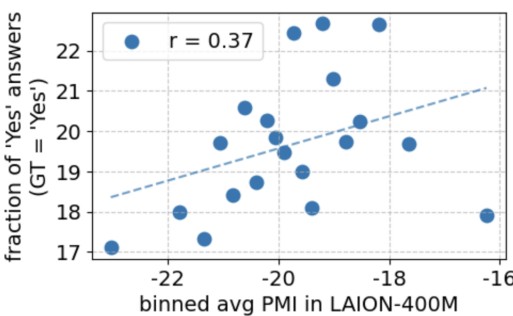

Figure 11: LMM bias toward answering 'yes' to high PMI inputs is much reduced when visual input is replaced with noise ($r = 0.77$ in the unaltered VQA setup).

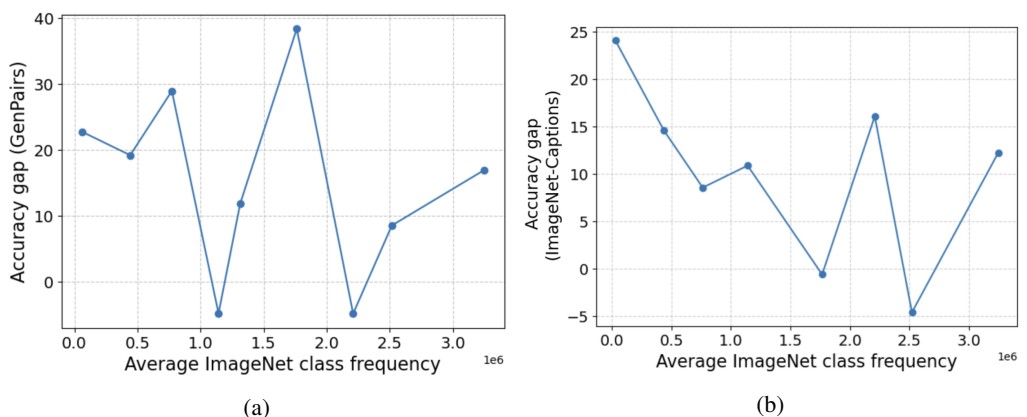

Figure 12: Little correlation between accuracy gap and frequency of ImageNet class in pretraining.

### B.3 ADDITIONAL RESULTS

#### B.3.1 CORRELATION WITH SINGLE CONCEPT FREQUENCY

To shed further light on the use of the PMI metric to decouple single concept frequencies with concept pair frequencies, we directly analyze the impact of single concept frequencies on accuracy gap (accuracy degradation due to low PMI inputs). In particular, *is there a certain amount of single concept data above which robustness to rare or unseen concept combinations emerges?* We explore this by correlating the frequency of an ImageNet class in LAION-400M, measured by Udandarao et al. (2024), with the zero-shot classification accuracy gap for concept pairs involving that class. Accuracy gap decreasing with increasing per-class frequency would suggest that simply collecting enough data is sufficient to bring about robustness. Our results in Figure 12 show minimal evidence that these quantities are related, indicating that scaling data collection on single concepts is not a reliable solution.

#### B.3.2 CORRELATION WITH NUMBER OF CONCEPTS

We additionally find that the number of concepts detected in each image does not have a clear correlation with accuracy gap (Figure 13). We use the concepts extracted following Udandarao et al. (2024) and the sizes of the error bars are inversely proportional to the number of examples in each bin.

### C COMPUTE REQUIREMENTS

We ran experiments on a combination of NVIDIA A100 and H100 GPUs. Non-trivial compute was needed for:

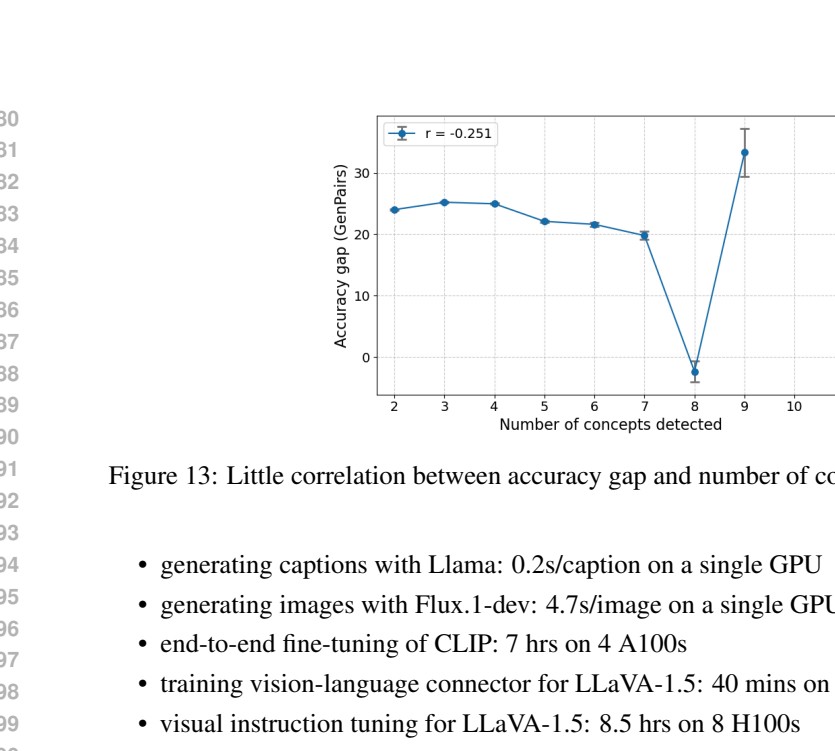

Figure 13: Little correlation between accuracy gap and number of concepts in the example.

- generating captions with Llama: 0.2s/caption on a single GPU
- generating images with Flux.1-dev: 4.7s/image on a single GPU
- end-to-end fine-tuning of CLIP: 7 hrs on 4 A100s
- training vision-language connector for LLaVA-1.5: 40 mins on 4 H100s
- visual instruction tuning for LLaVA-1.5: 8.5 hrs on 8 H100s

We estimate the total compute to be $\sim 1$ month of GPU time, including preliminary or failed experiments.

## D   LICENSES

- ImageNet (Deng et al., 2009) is licensed under BSD 3-Clause License.
- LAION (Schuhmann et al., 2021) is licensed under MIT License.
- TextVQA (Singh et al., 2019) is licensed under CC BY 4.0 License.
- VQAv2 (Singh et al., 2019) is licensed under CC BY 4.0 License.
- Flux.1-dev (Black Forest Labs, 2024) is under a Non-Commercial License.
- LLaVA (Liu et al., 2023a) is licensed under the Apache License 2.0.
- CLIP (Radford et al., 2021) and OpenCLIP (Cherti et al., 2023) are licensed under MIT License.

