# OpenReview forum: "Worse Together: Understanding the Brittleness of Multimodal Models on Rare Concept Pairs"
_ICLR.cc/2026/Conference — Submitted to ICLR 2026_

### Official Review · Reviewer_tMf7 · 2025-10-30

**Soundness:** 3
**Presentation:** 2
**Contribution:** 2
**Rating:** 4
**Confidence:** 3

**Summary:**

This paper conducts a series of empirical studies on how the co-occurrence of concepts affects the zero-shot performance of CLIP and LLaVA. It uses pointwise mutual information (PMI) to measure how likely two concepts are to be correlated in the pre-training datasets. Based on this, the authors create synthetic image datasets containing both high-PMI and low-PMI samples, and find an almost linear relationship between PMI values and zero-shot performance. This indicates that CLIP's predictions are not robust to the main concepts, but instead depend on this brittle relationship. The authors further use a low-PMI dataset to fine-tune CLIP, which increases zero-shot performance and transfers to other datasets. They propose that PMI-guided data curation can be used to improve robustness.

**Strengths:**

This paper presents a systematic study on the effects of the pre-training dataset. I find the strengths of this paper in the following aspects:

1. It establishes a linear relationship between PMI and zero-shot performance.

2. It introduces new datasets to investigate the robustness of CLIP models, and shows that fine-tuning on these datasets can transfer to other datasets.

3. It approaches the problem from a data perspective to address the dependency on co-occurring concepts.

**Weaknesses:**

1. Is this linear relationship between PMI and zero-shot performance dependent on the data curation? I mean, in line 185, does the selection threshold of 10,000 on frequency cause the linear relationship?

2. I am also confused about how Figures 2a and 2b are obtained. What does each dot represent? Do they correspond to a set of (c_accessory, c_ImageNet) concept pairs within a certain PMI range computed on LAION-400M, and then you test zero-shot performance on those concepts generated in GenPairs? Please provide a more detailed method explaining how the plots in your main results are produced.

3. The presentation needs improvement. For example, when I reached line 233, I still had no clear idea what a “key concept” is. Similarly, what is a “non-key concept pair”? These terms are not introduced before being used, which makes it hard for readers to follow.

4. Although I acknowledge that this paper provides a systematic study of the robustness of VLMs from a data perspective, the idea is not groundbreaking. Prior work on spurious correlations and fairness has similarly used counterfactual generation or disadvantaged-group data generation to improve robustness. The low-PMI-guided data construction shares a similar idea with these approaches.

**Questions:**

1. In lines 184–185, the authors describe how they extract concepts from LAION-400M. How do they avoid including non-noun or non-adjective words such as “a”, “the”, and “be”?

2. Since PMI between two concepts is symmetric, have you tried swapping the zero-shot prediction target to c_accessory? Would that produce a similar trend?

3. Flux.1-dev is a generative model, which seems to be tackling a harder task than zero-shot classification. I am curious why Flux.1-dev is able to generate low-PMI pairs, while CLIP struggles with them even in the zero-shot classification setting. I understand this question is hard to answer. I am asking mainly to discuss your perspective. Any insights or hypotheses would be helpful.

4. How does the fine-tuned model perform on concept pairs with high PMI? Does its performance decrease? What practical insight does this provide about curating training data for robustness? Since we cannot enumerate all low-PMI pairs in a dataset, what is the most efficient strategy?

**Details Of Ethics Concerns:**

I don't have any ethical concerns.

---

> ### Author Response · Authors · 2025-11-24
>
> We thank the reviewer for their time and insightful feedback. tMf7 notes that our paper **“presents a systematic study on the effects of the pre-training dataset”** by **“introduc[ing] new datasets to investigate the robustness of CLIP models, and show[ing] that fine-tuning on these datasets can transfer to other datasets”**. We answer specific questions below.
>
> > Is this linear relationship between PMI and zero-shot performance dependent on the data curation? I mean, in line 185, does the selection threshold of 10,000 on frequency cause the linear relationship?
>
> The thresholding was part of a previous revision of the paper and is no longer being applied (all concepts that pass the stopword filtration described in lines 188-192 are included). We thank the reviewer for pointing this out and have removed this from the current revision.
>
> > Although I acknowledge that this paper provides a systematic study of the robustness of VLMs from a data perspective, the idea is not groundbreaking. Prior work on spurious correlations and fairness has similarly used counterfactual generation or disadvantaged-group data generation to improve robustness. The low-PMI-guided data construction shares a similar idea with these approaches.
>
> While fine-tuning with disadvantaged data is an intuitive intervention, our analysis differs substantially from spurious correlations/fairness work in that groups are defined by single concept frequency, while we focus on the frequency of concept pairs. While previous work showed that the frequency of individual concepts in pretraining data is important for CLIP’s accuracy on those concepts, **we find that even for highly-represented concepts such as “broom”, pairing it with another common concept that rarely co-occurs with "broom" can cause a large drop in accuracy**. This interaction cannot be captured by a simple concept frequency/group-level analysis. In addition, our fine-tuning experiments provide empirical validation that fine-tuning with a simple data generation method (pasting in an accessory concept that has known PMI with the main image concept) produces improvements in both our zero-shot classification datasets, showing that robustness gains transfer as long as the concept pair distribution is sufficiently similar.
>
> > In lines 184–185, the authors describe how they extract concepts from LAION-400M. How do they avoid including non-noun or non-adjective words such as “a”, “the”, and “be”?
>
> We use the nltk library to specifically exclude “stop-words” such as “a”, “the”, etc. For the GenPairs data generating process, we have a more stringent filtration process involving part-of-speech tagging and LLM-assisted filtration of “visualizable” concepts (Sec A.1-A.2).
>
> > Since PMI between two concepts is symmetric, have you tried swapping the zero-shot prediction target to c_accessory? Would that produce a similar trend?
>
> We chose the ImageNet classes because the original CLIP paper already has a zero-shot classification recipe (e.g., templates for the text prediction target) tested specifically on ImageNet. Additionally, the ImageNet classes are chosen by humans and guaranteed to be “visualizable” concepts.
>
> > Flux.1-dev is a generative model, which seems to be tackling a harder task than zero-shot classification. I am curious why Flux.1-dev is able to generate low-PMI pairs, while CLIP struggles with them even in the zero-shot classification setting. I understand this question is hard to answer. I am asking mainly to discuss your perspective. Any insights or hypotheses would be helpful.
>
> In generative models, there is not necessarily optimization pressure to fuse representations between correlated concepts - in fact it may be more parsimonious to learn concepts individually and put them together. Prior work shows that the inductive bias in generative models seems amenable to this kind of interpolation between two seen objects/concepts ([2], which concludes that if the number of combinations in a training set goes above a certain threshold, the model starts to be able to generate novel combinations of seen concepts).
> Conversely, CLIP is trained to map visual and language input into the same embedding space and cluster similar images / text together - when concepts are correlated, CLIP may aggressively cluster the representations together to reduce the contrastive loss on frequent pairs.
>
> > How does the fine-tuned model perform on concept pairs with high PMI? Does its performance decrease?
>
> The performance in the highest PMI bin with finetuning (74.7%) is very similar to without finetuning (74.2%) (shown in Figure 3c).

---

> > ### Author Response · Authors · 2025-11-24
> >
> > > What practical insight does this provide about curating training data for robustness? Since we cannot enumerate all low-PMI pairs in a dataset, what is the most efficient strategy?
> >
> > While covering all concept combinations would be intractable, more deliberate effort can be made to collect data that is complementary to the natural distribution of internet data to provide diversity. Our findings suggest that increasing coverage of specific concept combinations—even selectively—improves accuracy on related combinations, highlighting the potential of more intentional data diversification.
> >
> > > I am also confused about how Figures 2a and 2b are obtained. What does each dot represent? Do they correspond to a set of (c_accessory, c_ImageNet) concept pairs within a certain PMI range computed on LAION-400M, and then you test zero-shot performance on those concepts generated in GenPairs? Please provide a more detailed method explaining how the plots in your main results are produced.
> >  ...
> > > The presentation needs improvement. For example, when I reached line 233, I still had no clear idea what a “key concept” is. Similarly, what is a “non-key concept pair”? These terms are not introduced before being used, which makes it hard for readers to follow.
> >
> > We thank the reviewer for these comments and have added text in lines 234-235 and 242-244 in the new revision to improve clarity.
> >
> > For Figures 2a/2b, we evaluate a CLIP ViT-B/32 pretrained on LAION-400M on zero-shot classification with the images in GenPairs and ImageNet-Captions (following the protocol defined in [1]).
> > For each example in each dataset, we calculate PMI for each valid concept pair, and take the average of these PMI values. Then we bin the PMI across examples into 20 equally sized bins (each of the 20 points in the plot), and show the correlation with classification accuracy.
> > We call the pair (c_accessory, c_ImageNet) the “key” concept pair, since the captions/images in GenPairs were explicitly generated from this concept pair.
> >
> > [1] Radford et al., 2021. Learning Transferable Visual Models From Natural Language Supervision.
> >
> > [2] Zhao et al., 2018. Bias and Generalization in Deep Generative Models: An Empirical Study.

---

> > > ### Comment · Reviewer_tMf7 · 2025-11-26
> > >
> > > The reviewer thanks you for your detailed response. It relieves my concerns about the counterfactual generation and provides more insight into the real application. I am open to raising my rating.

---

### Official Review · Reviewer_Sf9Q · 2025-10-31

**Soundness:** 2
**Presentation:** 3
**Contribution:** 2
**Rating:** 4
**Confidence:** 4

**Summary:**

This paper highlights a weakness of CLIP and LMMs: the performance is correlated to the Pointwise Mutual Information(PMI) of concept pairs in the training data. This work focusses on the failure of these models for rare combination of concepts, thus extending the current research being done on data-centric findings in vision-language. They also propose two benchmarks, GenPairs and ImageNet-Paste, for testing various concept pairs.

**Strengths:**

1. The study has a broad scope, which is very important. Data-centric studies should be applied to ALL model families that use them: hence extending the study beyond CLIP models to LMMs and from zero-shot classification to VQA is well-motivated.

2. Various experimental settings have been proposed, some novel. In addition to VQA and zero-shot classification, evaluation is done on GenPairs and ImageNet-Paste for multiple concepts. This is valuable for future research.

**Weaknesses:**

1. "While visual concepts can be difficult to define and extract from images": [1] has done exactly that. The concept extraction pipeline involves extracting visual concepts using an object tagger and provides a similar but more thorough text concept extraction function than this work. Additionally, [1] releases the concepts and artefacts for LAION-400M which is what the authors use. They could have just taken this metadata to run experiments. Having multimodal concepts allows for more fine-grained analysis, for example by taking the intersection of concepts per sample which ensures the concept exists in both image and text.

2. Here a concept is defined as single lemmatised words extracted from captions. This approach would not be able to distinguish polysemous words ("bat" the animal, "bat" the sports equipment). It also would not take into consideration multi-word concepts ("peanut butter").

3. Fine-grained PMI accuracy correlation experiments are essential but missing: it would be good to know how correlation varies when considering caption length, number of concepts in the caption, etc. Also one probe that would be interesting would be to correlate single frequency concept performance with pairwise performance (for example a high frequency concept paired with an extremely rare concept may outperform two low frequency concepts that are not that rare. A small ablation to show this could benefit the work).

4. I believe Eq 2 and 3 are incorrect: while estimating the empirical probability of a concept or pair of concepts in a dataset, it must be normalised by the number of documents not concepts (|D| instead of |C| and $\binom{|C|}{2}$). This affects the PMI calculation and also leads to asymmetric Laplace smoothing.

5. "models struggle to disentangle individual concepts and generalize to new combinations": the paper does not adequately establish this causality. Captions being unrelated/nosiy, synthetic image distribution bias in GenPairs, per-class frequency in pretraining could be factors that lead to that, not necessarily the co-occurence statistic.

6. Similar to point 5, the finding of LMMs being biased towards answering "yes" for high PMI inputs needs to be studied further. Specifically, does the bias come from the vision or text encoder? Recent works have shown that LMMs are over-reliant on the LLM component [2]. Suggested experiment: replace the image with noise and see if the answer is still "yes".

[1] Udandarao et al. No "Zero-Shot" Without Exponential Data: Pretraining Concept Frequency Determines Multimodal Model Performance, NeurIPS 2024

[2] Vo et al. Vision Language Models are Biased, 2025

**Questions:**

1. Why did the authors implement a frequency filter as high as 10,000? This would remove long-tailed concepts, which would skew the study of compositional performance, thus biasing the study of brittleness to rare concept pairs. Is it possible to check the number of concept pairs removed when setting a threshold (for example get different numbers for different thresholds)

2. Why have two different Laplace smoothing coefficients for single concept and multi-concept probabilities?

3. Fine-tuning CLIP with concept pairs with a wide range of PMI values shows promising avenues for data curation. Do the authors think this is good practise for pretraining or general tasks?

---

> ### Author Response · Authors · 2025-11-24
>
> We thank the reviewer for their time and insightful feedback. Sf9Q notes that our paper **“has a broad scope, which is very important”**, and our experimental settings are **“valuable for future research”**. We address specific concerns below.
>
> > W1: "While visual concepts can be difficult to define and extract from images": [1] has done exactly that. The concept extraction pipeline involves extracting visual concepts using an object tagger and provides a similar but more thorough text concept extraction function than this work.
>
>  **During the rebuttal period, we extracted concepts following [1] and find similar/stronger correlation between PMI and zero-shot classification accuracy compared to original results**. We ran both zero-shot classification experiments (ImageNet-Captions and GenPairs) with the concept extraction pipeline introduced in [1] and find that the more sophisticated concept extraction pipeline actually makes the PMI-accuracy correlation more pronounced. For ImageNet-Captions, we find Pearson r correlation between PMI and top-1 accuracy increased to 0.85 from 0.81, and r=0.82 -> 0.94 for top-5 accuracy. For GenPairs, r=0.97 for top-1 accuracy (unchanged from the original results), while r=0.96 -> 0.98 for top-5 accuracy. Further details are provided in Figure 10 and Section B.2.1. We thank the reviewer for this suggestion.
>
> > W2: Here a concept is defined as single lemmatised words extracted from captions. This approach would not be able to distinguish polysemous words ("bat" the animal, "bat" the sports equipment). It also would not take into consideration multi-word concepts ("peanut butter").
>
> Our results with concept extraction performed following [1] above uses concepts that incorporate polysemy, and leads to even better results (r=0.97 (unchanged) for GenPairs, r=0.81 -> 0.85 for ImageNet-Captions, see Figure 10 and Section B.2.1). [1] addresses polysemy by intersecting concepts extracted from the image and caption and defines some multi-word concepts in the image concept extraction pipeline.
>
> > W4: I believe Eq 2 and 3 are incorrect: while estimating the empirical probability of a concept or pair of concepts in a dataset, it must be normalised by the number of documents not concepts ($|\mathcal{D}|$ instead of $|\mathcal{C}|$ and $|\mathcal{C}| \choose 2$). This affects the PMI calculation and also leads to asymmetric Laplace smoothing.
>
> In our original results, we calculate PMI based on unigrams rather than document grouping of unigrams (e.g., if a word appears twice in a caption, it counts twice instead of once). We believe our Eq 2 and 3 are correct based on this definition of frequency, but we thank the reviewer for this comment and clarify this in the new revision (lines 153-155). We emphasize that we adopt the reviewer’s definition of PMI to produce Figure 10, since [1]’s published LAION400M concept frequencies are counted at the document level.
>
> > W3: Fine-grained PMI accuracy correlation experiments are essential but missing … one probe that would be interesting would be to correlate single frequency concept performance with pairwise performance (for example a high frequency concept paired with an extremely rare concept may outperform two low frequency concepts that are not that rare.)
>
> **On both GenPairs and ImageNet-Captions, we find little correlation (r=-0.24) between frequency of the ImageNet class concept and the accuracy gap** (difference in accuracy between highest and lowest PMI bins), reinforcing the idea that simply collecting more data about a concept does not reliably improve accuracy on rare or unseen concept pairs. We use the concepts extracted from [1] and the new PMI definition described above. We additionally look at accuracy vs number of concepts (shared b/t image and text concept extraction [1]) and also find a minimal correlation (r=-0.25) (with the caveat that very few examples have more >=8 shared concepts). We thank the reviewer for this suggestion and present additional results in Section B.3 of the new revision.

---

> ### Author Response · Authors · 2025-11-24
>
> > W5: "models struggle to disentangle individual concepts and generalize to new combinations": the paper does not adequately establish this causality. Captions being unrelated/nosiy, synthetic image distribution bias in GenPairs, per-class frequency in pretraining could be factors that lead to that, not necessarily the co-occurence statistic.
>
> We thank the reviewer for this comment and we certainly appreciate the complexity that the presence of all of these factors adds. However, we make an earnest effort to address each of these concerns:
> - **Captions being unrelated/noisy:** For our new results that extract concepts from images following [1], we only retain concepts that are detected in both the caption and the image itself, and found that this strengthened the correlation (Figure 10, Section B.2.1). In addition, we designed GenPairs to sidestep this issue by constructing images directly from text prompts that include the key concept pair, so the “captions” directly inform the contents of the image rather than the other way around. Our GenPairs results (Figures 2a, 2c) show very strong correlation between PMI and accuracy (r>=0.97 for top-1 accuracy).
> - **Synthetic image distribution bias in GenPairs:** We include evaluations on ImageNet-Captions, a subset of ImageNet, for zero-shot classification as well as VQA tasks that all leverage natural images. We consistently find a strong correlation between PMI and model accuracy across our natural and synthetic datasets.
> - **Per-class frequency in pretraining:** As described in our response to W3, we show during the rebuttal period that frequency of the ImageNet class in pretraining is not well correlated with accuracy gap (more data on a specific class does not improve robustness to inputs with low PMI concept pairs).
>
> > W6: Similar to point 5, the finding of LMMs being biased towards answering "yes" for high PMI inputs needs to be studied further. Specifically, does the bias come from the vision or text encoder? Recent works have shown that LMMs are over-reliant on the LLM component [2]. Suggested experiment: replace the image with noise and see if the answer is still "yes".
>
> A substantial part of the bias comes from the vision encoder, since we find a **substantially reduced correlation (r=0.37 compared to r=0.77) between bias towards answering ‘yes’ and PMI for the LM alone** (passing in noise as the visual input). This demonstrates that the LM’s bias does not sufficiently explain most of the correlation we observe with visual input. More details can be found in Section B.2.2 and Figure 11 of the new revision. We thank the reviewer for this suggestion.
>
> > Q1: Why did the authors implement a frequency filter as high as 10,000? This would remove long-tailed concepts, which would skew the study of compositional performance, thus biasing the study of brittleness to rare concept pairs. Is it possible to check the number of concept pairs removed when setting a threshold (for example get different numbers for different thresholds)
>
> The thresholding was part of a previous revision of the paper and is no longer being applied (all concepts that pass the stopword filtration described in lines 188-192 are included). We thank the reviewer for pointing this out and have removed this from the current revision.
>
> > Q2: Why have two different Laplace smoothing coefficients for single concept and multi-concept probabilities?
>
> Pair frequencies are on average 3 orders of magnitude lower than single concept frequencies, so we set the smoothing coefficient accordingly.
>
> > Q3: Fine-tuning CLIP with concept pairs with a wide range of PMI values shows promising avenues for data curation. Do the authors think this is good practise for pretraining or general tasks?
>
> We believe this would definitely be interesting to explore in the pretraining stage as well, we did not simply due to compute constraints.

---

### Official Review · Reviewer_HrCc · 2025-11-01

**Soundness:** 3
**Presentation:** 3
**Contribution:** 3
**Rating:** 6
**Confidence:** 4

**Summary:**

- This paper investigates how rare or unseen combinations of visual concepts affect the performance of CLIP and  large multimodal models.
- The author introduces a Pointwise Mutual Information–based framework to measure how often concept pairs appear together in training captions and finds that uncommon pairs lead to much lower accuracy.
- Experiments on both synthetic and real edited images show a strong correlation between concept-pair PMI and zero-shot accuracy of CLIP and MLLMs, suggesting that data composition has a stronger impact than model size.

**Strengths:**

- The motivation of the paper is clear and important.
- The paper introduces an interpretable, data-driven way to analyze multimodal robustness by framing model brittleness through concept-pair co-occurrence statistics.
- The analysis is supported by clear numerical evidence that convincingly demonstrate the impact of concept-pair rarity on performance.
- The study proposes a lightweight fine-tuning approach guided by PMI balancing, effectively improving generalization without extra data collection.

**Weaknesses:**

- The study heavily rely on caption word pairs to estimate co-occurrence may miss visual relationships that are not described in text.
- There’s no human analysis to confirm whether low-PMI failures align with human perception of concept rarity.

**Questions:**

- How can we tell if PMI differences cause performance drops rather than semantic bias?
- The experiment only cover CLIP and few MLLMs. What's the result on some closed-source models like GPT-4?
- Does increasing model size or data diversity reduce the PMI–accuracy correlation, and at what point does this relationship stop holding?
- How sensitive are the findings to the choice of tokenization?

---

> ### Author Response · Authors · 2025-11-24
>
> We thank the reviewer for their time and insightful feedback. HrCc notes that **“the motivation of the paper is clear and important”, and it “introduces an interpretable, data-driven”** framework whose results are **“supported by clear numerical evidence”**. We answer specific questions/concerns below.
>
> > The study heavily rely on caption word pairs to estimate co-occurrence may miss visual relationships that are not described in text.
>
>  During the rebuttal period, **we extracted concepts directly from images and find similar/stronger PMI-accuracy correlation compared to original results**. We ran both zero-shot classification experiments (ImageNet-Captions and GenPairs) with the concept extraction pipeline introduced in [1], which incorporates RAM++ to directly extract concepts from images. We find that the more sophisticated concept extraction pipeline actually makes the PMI-accuracy correlation more pronounced. For ImageNet-Captions, we find Pearson r correlation between PMI and top-1 accuracy increased to 0.85 from 0.81, and r=0.82 -> 0.94 for top-5 accuracy. For GenPairs, r=0.97 for top-1 accuracy (unchanged from the original results), while r=0.96 -> 0.98 for top-5 accuracy. Further details are provided in Figure 10 and Section B.2.1.
>
> > There’s no human analysis to confirm whether low-PMI failures align with human perception of concept rarity.
>
> We emphasize that this analysis is purely interested in understanding whether models will fail on concept pairs that co-occur infrequently _in the pretraining dataset_, which is an empirical measure of concept rarity for the model. Based on our results, as long as a concept pair is rare in pretraining, models are substantially more likely to fail on it even if humans perceive the concept pair to be common!
>
> > How can we tell if PMI differences cause performance drops rather than semantic bias?
>
> Our response assumes that by “semantic bias” the reviewer is referring to the possibility of an inductive bias from the model architecture towards certain concept pairs. While we did not ablate all parts of the model architecture, we tested a variety of model sizes on the CLIP zero-shot classification task (Figure 8) as well as LMM visual question answering tasks (Figure 9) and find that **PMI-accuracy correlation persists across the models we tested**. We note that the model size ablation study includes EVA01-g/14, which makes architectural/training recipe changes from vanilla ViT.
>
> > The experiment only cover CLIP and few MLLMs. What's the result on some closed-source models like GPT-4?
>
> While we did not test models like GPT-4, we did test LLaVA built on closed-source OpenAI CLIP in Figure 4 (right column). We found that PMI-accuracy correlation was very similar to that of LLaVA built on LAION400M-trained CLIP (e.g., r=0.76 for OpenAI CLIP evaluated on TextVQA compared to r=0.70 for LAION400M CLIP) even though OpenAI CLIP is not trained on LAION400M. We would like to run a study on GPT-4 and other closed-source models as soon as we can get funding to do so!
>
> > Does increasing model size or data diversity reduce the PMI–accuracy correlation, and at what point does this relationship stop holding?
>
> **Model size:** We performed a model size ablation for both the CLIP zero-shot classification task (Figure 8) as well as LMM visual question answering tasks (Figure 9), and find that PMI-accuracy correlation persists across the models we tested (up to ~500 GFLOPs).
>
> **Data diversity:** Our findings suggest that increasing coverage of specific concept combinations improves accuracy on related combinations, highlighting the potential of more intentional data diversification. While covering all concept combinations would be intractable, more deliberate effort can be made to collect data that is complementary to the natural distribution of internet data to provide diversity.
>
> > How sensitive are the findings to the choice of tokenization?
>
> While we did not directly ablate tokenization strategy, we experimented with the way our concepts are chosen/represented by directly extracting concepts from images [1], which better accounts for polysemy, and found a similar/stronger PMI-accuracy correlation (r=0.97 (unchanged) for GenPairs, r=0.81 -> 0.85 for ImageNet-Captions, see Figure 10 and Section B.2.1).
>
> [1] Udandarao et al. No "Zero-Shot" Without Exponential Data: Pretraining Concept Frequency Determines Multimodal Model Performance, NeurIPS 2024.

---

### Official Review · Reviewer_1F1L · 2025-11-01

**Soundness:** 2
**Presentation:** 3
**Contribution:** 3
**Rating:** 6
**Confidence:** 3

**Summary:**

This paper investigates how the co-occurrence statistics of concept pairs in CLIP’s pretraining data affect model robustness. The authors show a strong correlation between PMI of concept pairs and zero-shot accuracy, revealing that CLIP and LMMs struggle with rare or unseen concept combinations. They further demonstrate that fine-tuning with low-PMI (rare) pairs improves robustness and transfers to datasets. The paper highlights a key source of brittleness in multimodal models and provides a promising remedy.

**Strengths:**

- The paper presents a compelling study on how the co-occurrence statistics of concept pairs during pretraining affect the robustness of multimodal models such as CLIP and LMMs. The idea of quantifying this relationship through PMI is intuitive.

- The paper makes several interesting and valuable findings, including the strong correlation between PMI and zero-shot accuracy, the observation that fine-tuning with rare concepts can improve robustness and transfer to other datasets, and that LMMs exhibit similar patterns. These findings not only reveal the brittleness of current multimodal systems, but also provide insights into how to potentially mitigate it.

- They conducted many experiments to show strong empirical support for the claims.

**Weaknesses:**

- Regarding the GenPairs dataset construction, the generated images sometimes contain objects beyond the two intended concepts (e.g., in Figure 6, the image for the pair (coral, chambered nautilus) also includes other elements like a moon, and similarly for other images). Additionally, one could argue that those generated images with lower PMI values tend to have unusual backgrounds or environments as seen from the example images. These introduce potential confounders. The study does not fully control for these factors and therefore does not purely isolate the effect of PMI. So I’m not entirely sure if it is ideal to rely on these generated images for the main analysis.

- The results in Wiedemer et al. (2025) appear to have substantial overlap with this work. How this paper distinguishes itself from that prior study is not sufficiently discussed.

**Questions:**

Please see the questions raised in the Weaknesses section.

---

> ### Author Response · Authors · 2025-11-24
>
> We thank the reviewer for their time and insightful feedback. 1F1L notes that **“the paper makes several interesting and valuable findings” with “strong empirical support for the claims”** that **“not only reveal the brittleness of current multimodal systems, but also provide insights into how to potentially mitigate it”**.
> We answer specific questions/concerns below.
>
> > Regarding the GenPairs dataset construction, the generated images sometimes contain objects beyond the two intended concepts (e.g., in Figure 6, the image for the pair (coral, chambered nautilus) also includes other elements like a moon, and similarly for other images).
>
> We tested calculating PMI with all concept pairs present in the caption as well as PMI of the single intended concept pair and find very strong correlations with zero-shot accuracy for both (**r=0.99 for PMI on all concept pairs, r=0.97 for PMI on the intended concept pair**). Figures 2a (PMI on all pairs) and 2c (PMI on intended concept pair) show these results for GenPairs.
>
> > Additionally, one could argue that those **generated images with lower PMI values tend to have unusual backgrounds** or environments as seen from the example images. These introduce potential confounders. The study does not fully control for these factors and therefore does not purely isolate the effect of PMI. So I’m **not entirely sure if it is ideal to rely on these generated images** for the main analysis.
>
> Beyond the generated data, **we also include evaluations on existing real (non-generated) datasets**: ImageNet-Captions (a 440k subset of the ImageNet training dataset paired with their original Flickr captions, from [1], results in Figure 2b) and the VQA datasets in Section 6. In these cases, we still see very strong correlations between PMI and model response accuracy (e.g., r=0.81 for zero-shot classification on ImageNet-Captions).
>
> > The results in Wiedemer et al. (2025) appear to have substantial overlap with this work. How this paper distinguishes itself from that prior study is not sufficiently discussed.
>
> Unlike our work, Wiedemer et al. (2025) lacks evaluation on CLIP-based LMMs and does not take into account the frequency of concept combinations, which is central to our results. We go further to test on CLIP-based LMMs and show that the PMI-dependent behavior we find in CLIP persists in this setting, despite the incorporation of the LLM and additional training. Also, for known combinations (i.e., combinations where all objects o_1,...,o_n do appear together in pretraining) they ignore the frequency of the combination and only focus on the frequencies of the individual concepts. We argue that this doesn’t tell the whole story and **introduce the PMI metric**, which allows us to take into account the pretraining frequency of the combination of concepts as well as the frequencies of the individual concepts. This enables us to demonstrate that **performance is heavily dependent on the frequency of concept combinations rather than the individual concepts**, suggesting that simply collecting enough data about single concepts is not sufficient for robustness. We thank the reviewer for this suggestion and add a short discussion to the paper (lines 457-460).

---

### Author Response · Authors · 2025-11-24

We are grateful to the reviewers for their thoughtful and constructive feedback. We are glad that the reviewers found our paper well-motivated [HrCc, Sf9Q], systematic [tMf7], and its contributions valuable [1F1L, Sf9Q]. We have updated our submission to incorporate their feedback, which we believe has strengthened our submission. Below we summarize major additions:

- **Concept extraction** [HrCc, Sf9Q]: To address concerns regarding concept extraction from noisy text captions, we intersect text concepts with concepts extracted directly from images and found a similar/stronger PMI-accuracy correlation compared to original results. For ImageNet-Captions, we find Pearson r correlation between PMI and top-1 accuracy increased to 0.85 from 0.81, and r=0.82 -> 0.94 for top-5 accuracy. For GenPairs, r=0.97 for top-1 accuracy (unchanged from the original results), while r=0.96 -> 0.98 for top-5 accuracy. Further details are provided in Figure 10 and Section B.2.1.
- **Additional PMI-accuracy correlation results** [Sf9Q]: We find that per-class (single concept) frequency is not strongly correlated with accuracy gap, reinforcing the idea that simply collecting more data about a concept does not reliably improve accuracy on rare or unseen concept pairs. We also find substantially reduced correlation (r=0.37 compared to r=0.77 with visual input) between LMMs’ “yes” bias and PMI when testing the LM alone (passing in noise as the visual input), strengthening our conclusion that a substantial part of our observed bias comes from the vision encoder. Further details/figures are provided in Section B.2.2 and B.3.

We invite the reviewers to examine the updated submission and look forward to further discussion. If you wish to see the tracked changes, you can view the anonymized pdf linked here: https://anonymous.4open.science/r/rebuttal-8BD3/diff.pdf. We also respond to each reviewer individually below.

---

### Meta-Review · Area_Chair_tocM · 2026-01-08

**Summary:**

This paper claims that the frequency of co-occurrence of concept pairs in the pre-training data is a key factor for the zero-shot performance of multimodal models. The authors propose a PMI-based framework to measure the degree of co-occurrence for concept pairs and show that fine-tuning with low-PMI pairs can improve robustness and transferability to other datasets.

I agree with the following concerns raised by the reviewers:
- The contribution of the concept extraction framework may be less significant because:
  - Reviewer Sf9Q pointed out a highly relevant prior work that already proposes a method for concept extraction, and the authors confirmed that applying those concepts leads to the same conclusions.
  - The proposed concept pairs are model-extracted rather than human-annotated; therefore, some level of human evaluation seems necessary, as suggested by Reviewer HrCc.
- The conclusions share similarities with the claims of spurious correlations, as pointed out by Reviewer tMf7. Both suggest that the co-occurrence of concept pairs, or feature–label concept pairs, can lead to poor performance on rare cases.

Given this, I recommend a borderline reject.

**Reviewer Concerns:**

- (Not addressed) The contribution of the concept extraction framework may be limited, as concepts from prior work (e.g., Vo et al.) could be directly used.
- (Not addressed) Human evaluation results may be necessary.
- (Not addressed) The conclusions share similarities with claims about spurious correlations. The authors’ rebuttal does not address the point that spurious correlations can also be viewed as the frequency of concept pairs (feature–label concepts).
- (Addressed) There were concerns about images containing more than two concepts or captions not covering all concepts. These issues were properly explained by the authors during the rebuttal.
- (Addressed) Other clarification questions were properly addressed.

**Reviewer Scores:**

I believe Reviewer 1F1L, HrCc, Sf9Q will keep the score. Reviewer tMf7 mentions open to raising my rating.

---

### Decision · Program_Chairs · 2026-01-26

Reject